behaviour/ecology

geolocation, austral winter distribution, sea-ice concentration, iceberg, activity pattern, lunar cycle

**Author for correspondence:**
K. Delord
e-mail: karine.delord@cebc.cnrs.fr

# Antarctic petrels 'on the ice rocks': wintering strategy of an Antarctic seabird

K. Delord[1], A. Kato[1], A. Tarroux[2,3], F. Orgeret[1,4], C. Cotté[5,6], Y. Ropert-Coudert[1], Y. Cherel[1] and S. Descamps[3]

[1]Centre d'Etudes Biologiques de Chizé, UMR 7372 du CNRS-La Rochelle Université, 79360 Villiers-en-Bois, France
[2]Norwegian Institute for Nature Research, Fram Centre, 9296 Tromsø, Norway
[3]Norwegian Polar Institute, Fram Centre, 9296 Tromsø, Norway
[4]Department of Zoology, Nelson Mandela, University, Port Elizabeth, South Africa
[5]Laboratoire d'Océanographie et du Climat, Expérimentation et Approches Numériques, Institut Pierre Simon Laplace, Université Pierre et Marie Curie, Centre National de la Recherche Scientifique, Paris, France
[6]Sorbonne Universités (UPMC, Univ Paris 06)-CNRS-IRD-MNHN, LOCEAN Laboratory, Paris, France

KD, 0000-0001-6720-951X; AK, 0000-0002-8947-3634;
AT, 0000-0001-8306-6694; FO, 0000-0002-1940-7797;
CC, 0000-0002-2307-6435; YR-C, 0000-0001-6494-5300;
YC, 0000-0001-9469-9489; SD, 0000-0003-0590-9013

There is a paucity of information on the foraging ecology, especially individual use of sea-ice features and icebergs, over the non-breeding season in many seabird species. Using geolocators and stable isotopes, we defined the movements, distribution and diet of adult Antarctic petrels *Thalassoica antarctica* from the largest known breeding colony, the inland Svarthamaren, Antarctica. More specifically, we examined how sea-ice concentration and free-drifting icebergs affect the distribution of Antarctic petrels. After breeding, birds moved north to the marginal ice zone (MIZ) in the Weddell sector of the Southern Ocean, following its northward extension during freeze-up in April, and they wintered there in April–August. There, the birds stayed predominantly out of the water (60–80% of the time) suggesting they use icebergs as platforms to stand on and/or to rest. Feather $\delta^{15}$N values encompassed one full trophic level, indicating that birds fed on various proportions of crustaceans and fish/squid, most likely Antarctic krill *Euphausia superba* and the myctophid fish *Electrona antarctica* and/or the squid *Psychroteuthis glacialis*. Birds showed strong affinity for the open waters of the northern boundary of the MIZ, an important iceberg transit area, which offers roosting opportunities and rich prey fields. The strong association of Antarctic petrels with sea-ice cycle and icebergs suggests the species can serve, year-round, as a sentinel of environmental changes for this remote region.

# 1. Introduction

Unlike the Arctic and its constantly decreasing sea-ice extent, Antarctica experienced contrasting trends over the last three decades, with record maximum and minimum sea-ice extent within a few years [1–3]. Recently, the winter sea ice in the Antarctic Peninsula [4] and Weddell Sea region has been decreasing [5,6], coinciding with higher rates of iceberg calving and with large consequences for ecosystems associated with sea ice [7,8]. Marine predators using the pack-ice zone, polynyas or icebergs are thus expected to be impacted by these changes [9–13]. Among those, the Antarctic petrel (*Thalassoica antarctica*) is a wide-ranging species and year-round resident of Antarctic waters [14]. Antarctic petrels generally forage in close association with sea ice, cold water-masses and icebergs [9,14–18], where they capture primarily pelagic fish and crustaceans [19,20]. As such, any change in the icescape may have immediate consequences for petrel demography [21] and probably for their survival rate, as has been shown in other seabird species [22]. In this context, the non-breeding season, spanning several months of the austral winter, may constitute a critical period affecting population dynamics through an effect on individual survival [23]. Understanding how changes in the cryosphere during the non-breeding season affect vital rates is thus important, especially because of a general paucity of biological information during the winter months for Antarctic species. The relationships between seabirds and the sea-ice edge/iceberg habitats in Antarctica were examined in the past using at-sea observation data [10,24,25]. It was supported that physical rather than biological variables affect species assemblages [10,25], highlighting the effect of pack-ice zone (sea-ice edge and pack-ice extent) on the occurrence of Antarctic petrels. Yet, at-sea observations preclude linking ice habitats to individual strategy. Here, using 2 years of individual, longitudinal tracking data and stable isotopes, we examined the foraging ecology of Antarctic petrels over the non-breeding season and investigated how physical factors, mainly the sea-ice and icebergs affect birds distribution during this period.

# 2. Material and methods

Two types of miniature geolocating loggers (GLS; MK4083, Biotrack, UK, during the second season only, and LAT2500, Lotek, Canada, during the two seasons, less than 1% of the bird body mass, electronic supplementary material) were deployed on a total of 86 Antarctic petrels breeding at the world's largest known colony (Svarthamaren, Dronning Maud Land, Antarctica, 71°53′ S, 5°10′ E) in December 2011 ($n = 30$) and 2012 ($n = 56$). In the subsequent breeding seasons, returning birds were recaptured at their nests to retrieve their device and three to six body feathers were sampled from the lower back of most individuals. Sixteen birds were tracked two winter seasons in a row, i.e. they were instrumented in 2011, recaptured in 2012, re-instrumented in 2012 and recaptured again in 2013. Feather stable isotope values ($\delta^{13}$C and $\delta^{15}$N) were determined after recapture of the birds carrying GLS. Feathers reflect the diet at the time they were grown, because keratin is inert after synthesis [26–28]. In Antarctic fulmarine petrels, body moult is a gradual process extending over at least four months. It begins during late incubation, but most body feathers grow in the weeks following the completion of breeding [9,29]. Since the precise timing of synthesis of a given body feather was not known, isotopic measurements were performed on three to six fully grown feathers per bird. It is likely that most body feathers reflected dietary information about the previous inter-breeding period, i.e. moult, corresponding to the GLS tracking period [30]. It is important to note that we never observed any moulting feather on any of the handled birds during the breeding season, which suggests that Antarctic petrels in our study area start their moulting process later in the breeding season or after. For further details on the stable isotope method on body feathers and the moulting pattern of Antarctic petrels see the electronic supplementary material.

Geolocators provided light-level data used to calculate geographical positions (for procedures, see [20]). We restricted the analysis of the data from 15 April to 31 August in 2012 and 2013 to focus on the core period of the non-breeding season and exclude the equinoxes when latitude cannot be calculated. Distances to the breeding colony and to sea-ice edge (less than 15% in sea-ice concentration (SIC)) were calculated at each location. The geolocators also recorded saltwater immersion (wet/dry), providing data on birds' activity (on water or on ice/flying). Two types of devices were used (see electronic supplementary material) and for activity analysis we used only data from Biotrack devices from 2013, as they record cumulative time spent in each activity. More precisely, the Biotrack devices test for immersion in seawater every 3 s and record the proportion of time spent in each activity every 10 min. Although we do not know the sequences or the duration of the wet and dry periods during those 10 min, the devices still give a more complete image of the activity than the Lotek loggers would. The effect of moon (lunar phase and moon illumination) on activity pattern was examined using general linear mixed-effects models (see electronic supplementary material for further details).

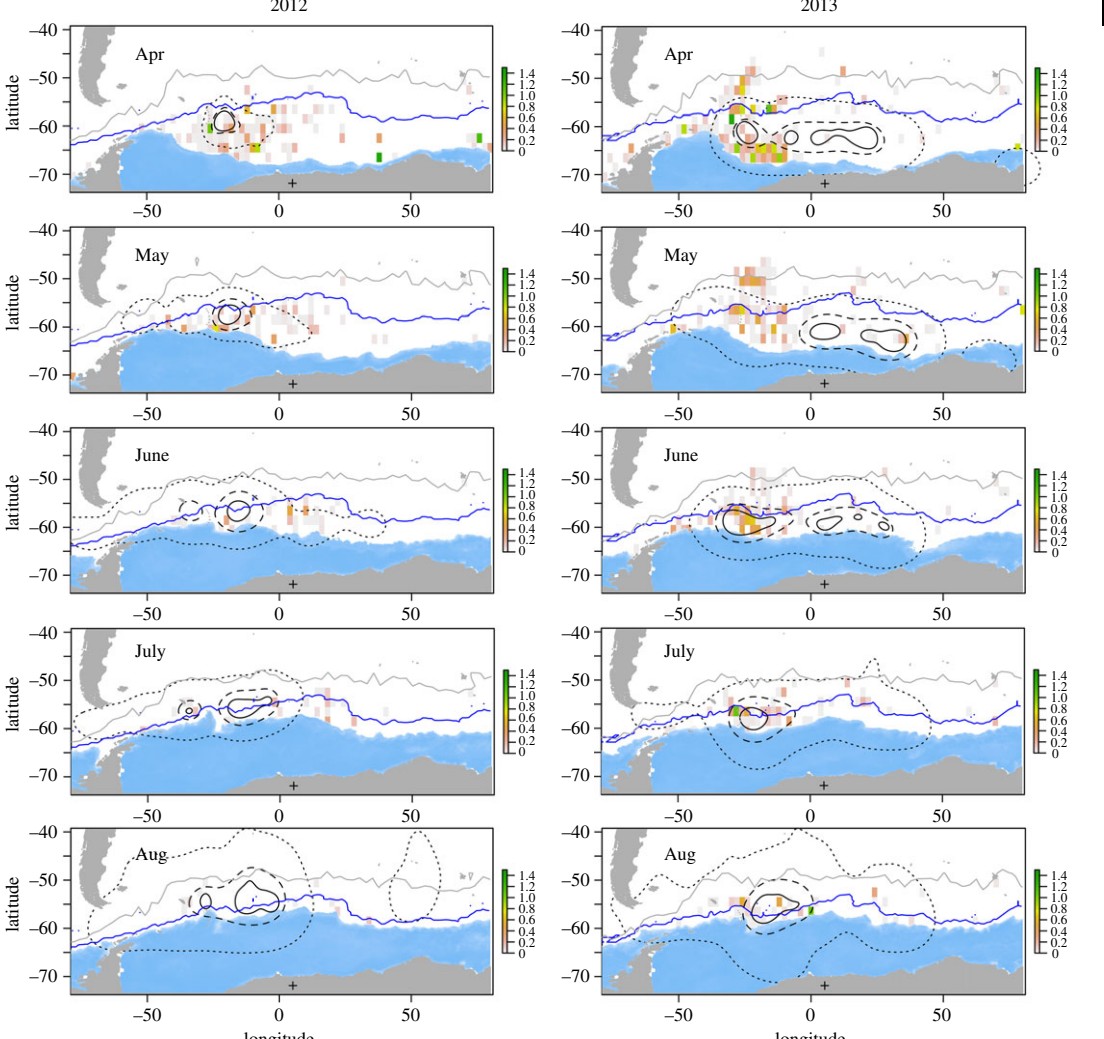

**Figure 1.** Monthly distribution of Antarctic petrels during the non-breeding period with surface area of small icebergs (in km², colour-coded according to a green-pink scale), SIC (blue) and maximum sea-ice extent (dark blue line). Polygons depict 95% (dotted), 50% (dashed) and 30% (solid) at-sea utilization distributions of petrels from the Svarthamaren breeding colony (black cross), Dronning Maud Land, Antarctica. Grey line shows the monthly position of the Polar Front (SHH = −0.58 m, [33]).

Because sea ice influences the behaviour of Antarctic petrels [17] and after considering collinearity among variables (electronic supplementary material, table S1), only four environmental covariates were kept to explain variations in the distribution of birds. SIC, the area covered by small icebergs (sm.ice, icebergs less than 3 km, in km²) and the residence time of large icebergs (la.ice, icebergs greater than 5 km, in days), as well as the sea surface height (SSH, as a proxy to define the location of fronts according to contour methods, introduced in [31]) were extracted and calculated as gridded ($2° \times 2°$ cell resolution) monthly values from April to August in 2012 and 2013 (electronic supplementary material). The effect of the covariates on the presence/absence of birds in a given cell was examined using generalized additive mixed-effects models (GAMMs), applying a separate smoothing function to each predictor variable (see supplementary material for statistical details). Generalized linear mixed-effects models were used to test the effects of distance to sea-ice edge and distance to colony. All tests were conducted in R v. 3.5.2 [32].

## 3. Results

A total of 69 bird-seasons were recovered over the two seasons, of which 64 yielded exploitable data ($n = 25$ bird-seasons in 2012, all devices from Lotek; $n = 39$ bird-seasons in 2013, 23 and 16 from Biotrack and Lotek, respectively) for a grand total of 48 individuals instrumented. Over the austral winter months, Antarctic petrels generally followed the expansion and subsequent recession of the sea-ice, being closest to the sea-ice edge around the midwinter (figures 1 and 2a). Although birds went farther away

**Figure 2.** (*a*) Average distance (±s.d.) of Antarctic petrels foraging to the north of the breeding colony (black and grey circles for 2012 and 2013, respectively) and to the north of the sea-ice edge (dark grey triangles). Distance to sea-ice edge was pooled for the 2 years as there was no significant difference between the years (see electronic supplementary material, table S2). (*b*) Proportion (±s.e.) of time Antarctic petrels spent in a dry state (out of water) by months over 2013 during daytime (open circles) and at night (closed circles). (*c*) Estimated smoothing curves (±s.e.) for environmental covariates in relation with the presence probability of petrels. Covariates considered are small (size of icebergs less than 3 km long) and large (residence time of icebergs greater than 5 km long) icebergs, sea-ice concentration (SIC, %) and sea surface height (SSH, in m). Months and years were used as fixed effects, and individuals as random effect.

from the colony in 2012 than in 2013, the distance between the birds and the ice edge did not change across years (electronic supplementary material, tables S2 and S3). Antarctic petrels travelled up to 3000 km from the colony (figure 2*a*; electronic supplementary material, tables S2 and S3) and the average distance to the north of the sea-ice edge stayed relatively constant (around 500 km) throughout the winter (figure 2*a*). In this wintering area, petrels spent on average around 60% of their daytime in a dry state (either in flight or on the ice, figure 2*b*), with this proportion increasing to 80% during the night. The activity of birds was found not to be influenced by lunar cycle during austral winter (electronic supplementary material, tables S5–S11; figure S1). GAMMs showed the importance of all the variables considered here (SIC, icebergs and SSH) in explaining the distribution of petrels (electronic supplementary material, table S4). In particular, Antarctic petrels always preferred 'open-water' zones, where SIC is less than 15%, and the probability of presence of the birds was above 0.5 when small icebergs (less than 3 km, figure 2*c*) were present and significantly increased with iceberg sizes, especially in June (electronic supplementary material, figure S2 and table S4; table 1). For large icebergs, although there was a statistical trend, we found a bimodal pattern that was difficult to interpret (figure 2*c*). Finally, presence probability of Antarctic petrels decreased drastically for SSH above a range of −0.6 to −0.3 m (roughly corresponding to the Polar Front) and birds were not recorded where SSH exceeded +0.6 m (Subtropical Front), indicating that they stayed and foraged almost exclusively within the Antarctic Zone, south of the Polar Front (table 1 and figures 1 and 2*c*; electronic supplementary material, figure S2).

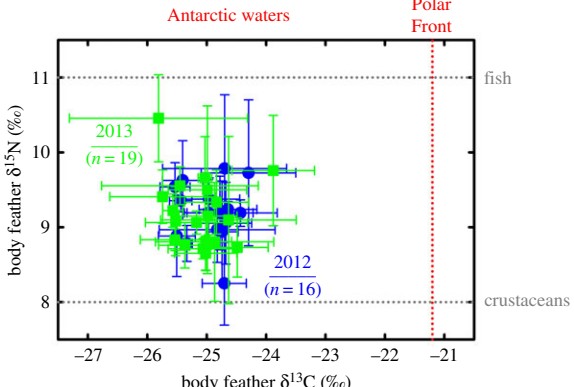

**Figure 3.** Feather $\delta^{15}$N versus $\delta^{13}$C values of Antarctic petrels from the inland colony of Svarthamaren in 2012 (blue) and 2013 (green). Values are means ± s.d. with four body feathers per individual bird. Vertical dotted line in red represents the $\delta13$C estimate of the Polar Front [34]. The two horizontal dotted lines in grey correspond to the $\delta^{15}$N estimates of a fish-based diet (upper line) and a crustacean-based diet (lower line), as measured in the feathers of chicks from the crustacean-eater chinstrap penguin *Pygoscelis antarctica* in [35] and the fish-eater king penguin *Aptenodytes patagonicus* in [36] (see the electronic supplementary material).

**Table 1.** Results of the GAMM explaining the presence/absence of Antarctic petrels as a function of environmental covariates. Variables selected in the best model and reference value occurrences are 2012 and April, respectively. The model explained 26.9% of the deviance.

| | term | estimate | s.e | statistic | *p*-value |
|---|---|---|---|---|---|
| parametric coefficients | intercept | −9.74 | 0.37 | −26.44 | <0.001 |
| | 2013 | 0.12 | 0.05 | 2.66 | <0.01 |
| | May | 0.82 | 0.05 | 16.08 | <0.001 |
| | June | 1.47 | 0.05 | 28.72 | <0.001 |
| | July | 1.91 | 0.05 | 35.28 | <0.001 |
| | August | 2.83 | 0.05 | 54.04 | <0.001 |
| | **term** | **edf** | **ref.df** | **statistic** | ***p*-value** |
| smooth terms | s(small icebergs) | 3.28 | 3.60 | 166.60 | <0.001 |
| | s(large icebergs) | 3.91 | 3.99 | 105.34 | <0.001 |
| | s(sea ice concentration) | 3.95 | 4.00 | 2523.92 | <0.001 |
| | s(sea surface height) | 3.88 | 3.99 | 6257.35 | <0.001 |
| | s(bird ID) | 41.42 | 45.00 | 856.54 | <0.001 |

Body feather $\delta^{13}$C and $\delta^{15}$N values ($n = 139$) averaged −25.0 ± 0.7 and 9.2 ± 0.6‰, respectively. They encompassed large ranges, amounting to differences of 4.0‰ (from −27.1 to −23.1‰) and 3.4‰ (from 7.8 to 11.2‰). At the individual level ($n = 16$ and 19 in 2012 and 2013, respectively), $\delta^{13}$C and $\delta^{15}$N values ranged from −25.8 ± 1.5 to −23.9 ± 0.7‰, and from 8.2 ± 0.6 to 10.5 ± 0.6‰, respectively (figure 3).

## 4. Discussion

During summer, Antarctic petrels track the phenology of the sea ice over the breeding season, i.e. they target ice-melting areas as preferred foraging grounds [9]. Here, we show that this tight link with sea ice extends throughout the non-breeding period, with birds tracking the phenology of the sea-ice advance foraging primarily in open waters associated with icebergs. This seems to be in line with the fact that food availability at the surface appeared not to decrease in pack ice (contrary to open water) during winter [37,38].

Our results demonstrate the role that the marginal ice zone (MIZ, see [39])—a dynamic and biologically active region that transitions from the dense inner pack-ice zone to ice-free open ocean

(e.g. [40,41])—plays in driving the distribution of Antarctic petrels during the winter months, as has been reported for other seabirds in the Arctic [42,43]. Birds remain within a large belt that stretches from the northern physical edge of the pack ice defined by satellite measurement (open water being defined here and elsewhere as cells where SIC was less than 15%; [44]) to the Polar Front in the North. This foraging zone included the progressively narrowing seasonal ice zone (SIZ) in the south and the permanent open ocean zone (POOZ) in the north. Stable isotopes confirmed this pattern, with all $\delta^{13}$C values being highly negative, indicating foraging exclusively south of the Polar Front during the body feather moulting period. The single dietary study on Antarctic petrels during winter highlighted the importance of two prey, the myctophid *Electrona antarctica* and the squid *Psychroteuthis glacialis* [45]. Those two organisms have similar high $\delta^{15}$N values that preclude differentiation [46,47], thus contrasting with the lower value of a third key prey species [19,20], the Antarctic krill *Euphausia superba* [46]. Feather $\delta^{15}$N values encompassed one full trophic level, meaning that prey ranged from crustacean (most likely *Eu. superba*) to fish/squid (*El. antarctica/P. glacialis*), with diet including various proportions of the two groups in most cases (figure 3). The behaviour of petrels was well known to be impacted by the lunar phase, through the influence on their activity level at breeding colonies or on their activity at sea during or outside the breeding period [48–51]. The lunar cycle is believed to strongly influence the vertical distribution of potential marine prey species, making them more accessible and/or visible to their predators [48,50]. Interestingly, and in contradiction with these references, the behaviour of Antarctic petrels was not affected by the lunar cycle at the fine temporal scale of our study, while the potential prey, namely myctophid fish and *Eu. superba*, both perform diel vertical migration and are known to be influenced by the moonlight. This contradiction requires further investigations.

The Polar Front coincides with the northern boundary of the iceberg occurrence [52]. It represents a highly dynamic and heterogeneous region where the ice platform breaks, creating zones of mixing and turmoil, but also a corridor for icebergs that originate mostly from the ice shelves in the Weddell Sea and drift with the Antarctic Circumpolar Current [53,54]. Interestingly, the activity data indicated that petrels were predominantly out of the water (60–80% of the time), to a degree similar to that recorded during the chick-rearing season, when birds spend extended periods of time at their nest between trips at sea [55]. The Antarctic petrels are frequently observed to rest and huddle on ice surfaces (floes and bergs) [10]. This strongly suggests that Antarctic petrels use icebergs as platforms to stand on and/or rest during winter, although it was not possible to confirm this with the coarse scale at which petrels were, by necessity, tracked in our study. The exact nature of the interaction between the petrels and the icebergs remains to be elucidated, but recent reports of icebergs being visited by a wide range of foraging seabirds, including Antarctic petrels [16,18], supports the view that these structures constitute important features shaping seabirds' distribution. Icebergs are associated with higher ocean net primary productivity, especially in the SIZ and POOZ, suggesting a role of hotspots of biological activity for these features [8,54,56–58]. This association is relevant in the context of an observed increase in iceberg occurrence over the last decades in the southern Atlantic section of the Southern Ocean, in conjunction with global warming [7,8,53,59,60].

## 5. Conclusion

The Antarctic petrel is associated throughout its annual cycle with icebergs and/or open waters that follow the large-scale seasonal sea-ice movements. We demonstrated that the birds were associated during the entire inter-breeding period with MIZ (icescape: pack-ice zone, sea-ice edge, cold water-masses and icebergs) and frontal system, confirming at-sea observations. Our individual tracking investigation allowed us to analyse processes involved at the individual level (temporal and spatial scale) on distribution and behaviour throughout the entire inter-breeding period. This brings new longitudinal insights on effects of environmental parameters on distribution during austral winter over a large oceanic area. The Antarctic petrel was previously found to forage mainly on myctophid fish and krill during the breeding period [10]. Feather stable isotope values suggest that birds also feed on krill and myctophids during the non-breeding period, so that it is reasonable to assume that the Antarctic petrel can be used as a good indicator of prey availability throughout the year. This association makes the species a relevant sentinel of environmental changes, like the closely related fulmarine snow petrel *Pagodroma nivea* [61], the emperor *Aptenodytes forsteri* [62] and the Adélie *Pygoscelis adeliae* [63] penguins do in other areas of the Antarctic. Hence, we encourage ecological studies to be conducted more broadly on this species, in various Antarctic locations, so as to help

provide key information on environmental changes to conservation bodies, e.g. the Commission for the Conservation of Antarctic Marine Living Resources (https://www.ccamlr.org).

Ethics. All capture and handling procedures were in accordance with the permits provided by the Norwegian Animal Research Authority (NARA/FDU permit nos. 3714 and 5746).

Data accessibility. Datasets supporting this article were uploaded as part of the electronic supplementary material.

Authors' contributions. Study design: K.D., S.D., A.T. and Y.C. Fieldwork: S.D., A.T. and Y.C. Data analysis and processing: K.D., A.K., C.C., F.O., A.T., Y.C. and K.D. wrote the text and all authors edited and revised the manuscript, gave final approval for publication and agreed to be held accountable for the content therein.

Competing interests. We declare we have no competing interests.

Funding. Funding was provided by the Norwegian research council (NARE programme). This study is a contribution to programme SENSEI (grant agreement no. 2017-00000006497 to C. Barbraud and Y.R.-C.) funded by the BNP Paribas Foundation.

Acknowledgements. We thank all the fieldworkers who contributed to collecting the data, R. Reisinger for help with statistical modelling and S. Harris for the revision of the English version. H. Lormée was an active helper in the CEBC ThinkTank morning sessions. The authors thank the anonymous referees for helpful reviews and constructive suggestions to improve the manuscript.

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
