## [Reviewer comments · Royal Society Open Science]

Review History

RSOS-191429.R0 (Original submission)

Review form: Reviewer 1

Is the manuscript scientifically sound in its present form?

Yes

Are the interpretations and conclusions justified by the results?

Yes

Is the language acceptable?

Yes

Do you have any ethical concerns with this paper?

No

Have you any concerns about statistical analyses in this paper?

No

Recommendation?

Accept as is

Comments to the Author(s)

I think the authors made a good job responding all the concerns and revising the manuscript.

So, I have only these minor comments, that can be also fixed at the proof stage

line 63: "tacking data" - repeated twice

lines 81-84 - the two sentences are somehow repeating

line 204: "good indicator or prey" - should be good indicator OF prey

Review form: Reviewer 2**Is the manuscript scientifically sound in its present form?**

No

Are the interpretations and conclusions justified by the results?

No

Is the language acceptable?

Yes

Do you have any ethical concerns with this paper?

No

Have you any concerns about statistical analyses in this paper?

Yes

Recommendation?

Major revision is needed (please make suggestions in comments)

Comments to the Author(s)

Antarctic petrels 'on the ice rocks': wintering strategy of an Antarctic seabird

It is an interesting paper that describes the non-breeding distribution and isotopic values of Antarctic petrels. Although the paper has potential, but I believe the authors need to improve it to achieve its full potential. Below I describe some suggestions that could help the authors improve their paper.

Abstract:

Please begin the abstract with a sentence explain why this study is important or which gap of knowledge do you want to fill in with your manuscript.

Line 31: It is not clear from your abstract (nor title) the importance of the stable isotope analysis in your study. If you just want to determine where are Antarctic petrel during the non-breeding period and their association with sea-ice concentration and icebergs, which extra information does stable isotope data give you? Explain it in the abstract.

Are isotopic values of these potential prey available in bibliography? If there is, it would be better if you perform isotopic diet reconstruction instead of putting here the name of the potential prey that you did not verify.

Line 38: You did not include anything regarding to the use of the activity data in the abstract, however it is an important part of your manuscript. I would suggest you mention it in the abstract.

Introduction

Lines 62-64: Did you compare the non-breeding area and isotopic values between the two study years at intraindividual level? Furthermore, if you want to check the consistency of the foraging ecology and how non-breeding distribution at individual level you should have more than 2 year-data.

Lines 63-64: tracking data is repeated twice

Material and methods

Sample sizes are not clearly explained.

Geolocators: You mention at the end of the introduction that you used 2-years of individual longitudinal tracking and isotopic data so you should have for each (or to the majority) of the individuals 2 years of geolocation and of stable isotopic data. You mention in methods that you deployed geolocators to 86 petrels and recovered 69 geolocators. How many individuals correspond this 69 geolocators? Do these geolocators have only 1 year of data or is there any with 2 years? So, did you track approximately 35 individuals for 2 years? How many geolocators of Biotrack and Lotek did you recovered from each year? You only used Biotrack geolocators for the activity data analysis, but I did not find a clear information of how many geolocation trips of each year you used for this analysis.

Feather sampling: based on the figure of stable isotope results, you have feathers from 16 individuals in one year and 19 from another year. So, in total you would have at maximum 16 individual with isotopic data from the 2 study years, correct? Why don't you have more feather samples since you recovered 64 geolocators with data?

Line 78: Did you sample the feathers during the incubation period? If you collected them during the chick-rearing did you select those feathers more wore to ensure that the feathers sampled were moulted in the previous year?

Line 88: Do you know the breeding success of your tracked individual? It is known that migratory phenology can differ between individuals with different breeding success, with failed breeders migrating earlier and, also, moulting earlier. Thus, failed breeders could moult a greater proportion of their body feathers still during the breeding period in contrast with successful breeders that may moult the majority already in their non-breeding areas.

Lines 92-96: Maybe you should mention that although Biotrack geolocation activity data have more resolution, you do not know the sequence or duration of the wet and dry periods during those 10-min. Please refer how many geolocators did you used for this analysis (and how many from different individuals)

Lines 96-98: Normally petrels' behaviour is influenced by the moonlight specially the ones relying on prey such as myctophids or krill that perform the diel vertical migration. Furthermore, during the winter in such latitudes the diel vertical migration can be influenced by the moonlight. So, I would suggest you check it the moonlight influence in petrels' activity in more detailed. (1) Please considered to use a GAMM similar for example to <https://www.int->

res.com/abstracts/esr/v40/p189-206/ instead of the GLMM to test the effect of the moon. In that paper the authors used a continuous variable of moon-phase instead of a categorical one. (2) Also considered to use the time spend on water instead of out of water, as I would expect that the relationship with moonlight would be greater in the time spent on water during night/dawn/dusk, because if there is moonlight birds can spend more time on water foraging (3) instead of categorized day or night use a continuous daylight variable or at least use 4 categories: day, night, dawn and dusk, because many seabirds forage during dawn and dusk and in the present manuscript is not clear if you include those periods in the daylight or in the night period.

Lines102-103: please refer the units of the residence time of large icebergs
Any particular reason why you used 30% and 70% UD kernels and not the most common ones of 50% and 95% generally representing the core-area and home-range distributions?
Wouldn't be possible to use the directly the time spent per cell grid in the GLMM and GAMM instead of converting it to presence/absence grids?

Results

Lines 115-116: please refer to the Table S2 and S3 at end of this sentence.

Lines 119-120: to try to understand if the bird was flight or resting on ice you could (1) check the duration of the continuously dry periods; large periods probably correspond to resting, while short periods probably correspond to flights, although this will be different among species if there is information about the maximum continuous flight of Antarctic petrels all dry periods above that thresholds would probably correspond to resting; (2) check in which period of the day there are long dry periods occur. In figure 2b you present the date pooled for each month making difficult to better understand the behaviour of your study species. It would be better if you check how the activity of Antarctic petrel change throughout the day (and maybe day and month) by plotting a GAMM like for example the Figure 1 of the article Ramos R., Ramírez I., Paiva V.H., Militão T., Biscoito M., Menezes D., Phillips R.A., Zino F., González-Solís J. (2016) Global spatial ecology of three closely-related gadfly petrels. *Scientific Reports* 6: 23447. Furthermore, you could calculate a grid of the "night flight index" (which in the case of your species the dry periods would be flight/resting) and compare using a GLMM with the grid of the small icebergs to check if near small icebergs your birds tend to spend more time on dry, which could suggest they are resting on those areas

Figure 2c (page 30): please include the units of the variables in the graphs or in the legend of the figure. Furthermore, in the legend you need to explain that the variable of small iceberg is representing the size of small icebergs until 3km of length, while the large iceberg is not the size but the residence time of large icebergs (>5km of length).

As far as I notice, you only refer in the legend of Figure 2 and not in anywhere else of the paper that the activity data is only from 2013. Please refer it in methods or in results.

Figure 3 (page 31): Are there isotopic values of the three main prey species of Antarctic petrel? If there is please perform an isotopic diet reconstruction, it is more precise than just put these lines and it will allow you to know (and discuss) the proportion of krill versus fish/squid. You should refer that the reference isotopic values of penguins you used are from chicks. Isotopic data from chicks may be biased regarding to adults due to metabolic process related with quick body growth. However, penguin adults moult while fasting which can also affect isotopic values. It would be better if you could find a reference isotopic values of the prey or a petrel species. If none of these not available, I agree it is better to include those of penguin feathers as you did.

Table S4: These are not deltaAIC, they are probably AIC values. Please rectify and present the AICweight too.

Some of the tables of the GLMM and GAMM results are lacking the degree of freedom.

Please include it in all the respective tables

Discussion

Line 169: Be careful as you may be contradicting yourself as in the abstract and part of the discussion you suggest that Antarctic petrels rely on myctophid fish and krill both performing diel vertical migration and probably influenced by the moonlight.

Decision letter (RSOS-191429.R0)

27-Nov-2019

Dear Dr Ropert-Coudert,

The editors assigned to your paper ("Antarctic petrels 'on the ice rocks': wintering strategy of an Antarctic seabird") have now received comments from reviewers. We would like you to revise your paper in accordance with the referee and Associate Editor suggestions which can be found below (not including confidential reports to the Editor). Please note this decision does not guarantee eventual acceptance.

Please submit a copy of your revised paper before 20-Dec-2019. Please note that the revision deadline will expire at 00.00am on this date. If we do not hear from you within this time then it will be assumed that the paper has been withdrawn. In exceptional circumstances, extensions may be possible if agreed with the Editorial Office in advance. We do not allow multiple rounds of revision so we urge you to make every effort to fully address all of the comments at this stage. If deemed necessary by the Editors, your manuscript will be sent back to one or more of the original reviewers for assessment. If the original reviewers are not available, we may invite new reviewers.

- Data accessibility

<http://datadryad.org/submit?journalID=RSOS&manu=RSOS-191429>

- Competing interests

- Authors' contributions

- Acknowledgements

- Funding statement

Best regards,

on behalf of the Associate Editor, and Professor Kevin Padian (Subject Editor)
openscience@royalsociety.org

Associate Editor's comments to Author:

Thank you for submitting this transfer to Royal Society Open Science. Two reviewers have assessed the paper, and based on their commentary, a revision is required to address a number of concerns raised.

Please note that, in most cases, only one round of revision will be permitted, and you should work to satisfy all the requirements identified by the reviewers. Please ensure you include a full point by point response when you resubmit a revised paper. It would be extremely helpful if you could highlight all revisions made within the manuscript document.

Good luck!

Reviewers' Comments to Author:

Reviewer: 1

Comments to the Author(s)

I think the authors made a good job responding all the concerns and revising the manuscript.

So, I have only these minor comments, that can be also fixed at the proof stage

line 63: "tacking data" - repeated twice

lines 81-84 - the two sentences are somehow repeating

line 204: "good indicator or prey" - should be good indicator OF prey

Reviewer: 2

Comments to the Author(s)

Antarctic petrels 'on the ice rocks': wintering strategy of an Antarctic seabird

It is an interesting paper that describes the non-breeding distribution and isotopic values of Antarctic petrels. Although the paper has potential, but I believe the authors need to improve it to achieve its full potential. Below I describe some suggestions that could help the authors improve their paper.

Abstract:

Please begin the abstract with a sentence explain why this study is important or which gap of knowledge do you want to fill in with your manuscript.

Line 31: It is not clear from your abstract (nor title) the importance of the stable isotope analysis in your study. If you just want to determine where are Antarctic petrel during the non-breeding period and their association with sea-ice concentration and icebergs, which extra information does stable isotope data give you? Explain it in the abstract.

Are isotopic values of these potential prey available in bibliography? If there is, it would be better if you perform isotopic diet reconstruction instead of putting here the name of the potential prey that you did not verify.

Line 38: You did not include anything regarding to the use of the activity data in the abstract,

however it is an important part of your manuscript. I would suggest you mention it in the abstract.

Introduction

Lines 62-64: Did you compare the non-breeding area and isotopic values between the two study years at intraindividual level? Furthermore, if you want to check the consistency of the foraging ecology and how non-breeding distribution at individual level you should have more than 2 year-data.

Lines 63-64: tracking data is repeated twice

Material and methods

Sample sizes are not clearly explained.

Geolocators: You mention at the end of the introduction that you used 2-years of individual longitudinal tracking and isotopic data so you should have for each (or to the majority) of the individuals 2 years of geolocation and of stable isotopic data. You mention in methods that you deployed geolocators to 86 petrels and recovered 69 geolocators. How many individuals correspond this 69 geolocators? Do these geolocators have only 1 year of data or is there any with 2 years? So, did you track approximately 35 individuals for 2 years? How many geolocators of Biotrack and Lotek did you recovered from each year? You only used Biotrack geolocators for the activity data analysis, but I did not find a clear information of how many geolocation trips of each year you used for this analysis.

Feather sampling: based on the figure of stable isotope results, you have feathers from 16 individuals in one year and 19 from another year. So, in total you would have at maximum 16 individual with isotopic data from the 2 study years, correct? Why don't you have more feather samples since you recovered 64 geolocators with data?

Line 78: Did you sample the feathers during the incubation period? If you collected them during the chick-rearing did you select those feathers more wore to ensure that the feathers sampled were moulted in the previous year?

Line 88: Do you know the breeding success of your tracked individual? It is known that migratory phenology can differ between individuals with different breeding success, with failed breeders migrating earlier and, also, moulting earlier. Thus, failed breeders could moult a greater proportion of their body feathers still during the breeding period in contrast with successful breeders that may moult the majority already in their non-breeding areas.

Lines 92-96: Maybe you should mention that although Biotrack geolocation activity data have more resolution, you do not know the sequence or duration of the wet and dry periods during those 10-min. Please refer how many geolocators did you used for this analysis (and how many from different individuals)

Lines 96-98: Normally petrels' behaviour is influenced by the moonlight specially the ones relying on prey such as myctophids or krill that perform the diel vertical migration. Furthermore, during the winter in such latitudes the diel vertical migration can be influenced by the moonlight. So, I would suggest you check it the moonlight influence in petrels' activity in more detailed. (1) Please considered to use a GAMM similar for example to <https://www.int-res.com/abstracts/esr/v40/p189-206/> instead of the GLMM to test the effect of the moon. In that paper the authors used a continuous variable of moon-phase instead of a categorical one. (2) Also considered to use the time spend on water instead of out of water, as I would expect that the relationship with moonlight would be greater in the time spent on water during night/dawn/dusk, because if there is moonlight birds can spend more time on water foraging (3)

instead of categorized day or night use a continuous daylight variable or at least use 4 categories: day, night, dawn and dusk, because many seabirds forage during dawn and dusk and in the present manuscript is not clear if you include those periods in the daylight or in the night period.

Lines 102-103: please refer the units of the residence time of large icebergs

Any particular reason why you used 30% and 70% UD kernels and not the most common ones of 50% and 95% generally representing the core-area and home-range distributions?

Wouldn't be possible to use the directly the time spent per cell grid in the GLMM and GAMM instead of converting it to presence/absence grids?

Results

Lines 115-116: please refer to the Table S2 and S3 at end of this sentence.

Lines 119-120: to try to understand if the bird was flight or resting on ice you could (1) check the duration of the continuously dry periods; large periods probably correspond to resting, while short periods probably correspond to flights, although this will be different among species if there is information about the maximum continuous flight of Antarctic petrels all dry periods above that thresholds would probably correspond to resting; (2) check in which period of the day there are long dry periods occur. In figure 2b you present the date pooled for each month making difficult to better understand the behaviour of your study species. It would be better if you check how the activity of Antarctic petrel change throughout the day (and maybe day and month) by plotting a GAMM like for example the Figure 1 of the article Ramos R., Ramírez I., Paiva V.H., Militão T., Biscoito M., Menezes D., Phillips R.A., Zino F., González-Solís J. (2016) Global spatial ecology of three closely-related gadfly petrels. *Scientific Reports* 6: 23447. Furthermore, you could calculate a grid of the "night flight index" (which in the case of your species the dry periods would be flight/resting) and compare using a GLMM with the grid of the small icebergs to check if near small icebergs your birds tend to spend more time on dry, which could suggest they are resting on those areas

Figure 2c (page 30): please include the units of the variables in the graphs or in the legend of the figure. Furthermore, in the legend you need to explain that the variable of small iceberg is representing the size of small icebergs until 3km of length, while the large iceberg is not the size but the residence time of large icebergs (>5km of length).

As far as I notice, you only refer in the legend of Figure 2 and not in anywhere else of the paper that the activity data is only from 2013. Please refer it in methods or in results.

Figure 3 (page 31): Are there isotopic values of the three main prey species of Antarctic petrel? If there is please perform an isotopic diet reconstruction, it is more precise than just put these lines and it will allow you to know (and discuss) the proportion of krill versus fish/squid. You should refer that the reference isotopic values of penguins you used are from chicks. Isotopic data from chicks may be biased regarding to adults due to metabolic process related with quick body growth. However, penguin adults moult while fasting which can also affect isotopic values. It would be better if you could find a reference isotopic values of the prey or a petrel species. If none of these not available, I agree it is better to include those of penguin feathers as you did.

Table S4: These are not deltaAIC, they are probably AIC values. Please rectify and present the AICweight too.

Some of the tables of the GLMM and GAMM results are lacking the degree of freedom.

Please include it in all the respective tables

Discussion

Line 169: Be careful as you may be contradicting yourself as in the abstract and part of the discussion you suggest that Antarctic petrels rely on myctophid fish and krill both performing diel vertical migration and probably influenced by the moonlight.

Author's Response to Decision Letter for (RSOS-191429.R0)

See Appendix A.

RSOS-191429.R1 (Revision)

Review form: Reviewer 2

Is the manuscript scientifically sound in its present form?

Yes

Are the interpretations and conclusions justified by the results?

Yes

Is the language acceptable?

Yes

Do you have any ethical concerns with this paper?

No

Have you any concerns about statistical analyses in this paper?

No

Recommendation?

Accept with minor revision (please list in comments)

Comments to the Author(s)

I believe you addressed most of my concerns and in those cases you did not address them you gave reasonable explanations.

Just a few minor suggestions:

- You refer in the response to my review that "it is important to note that we never observed any moulting feather on any of the handled birds during the breeding season, which suggests that Antarctic petrels in our study area start their moulting process later in the breeding season or after." I would suggest you include this information in page 14 line 43.
- please change the decimal symbol to dot in tables S6 onward.

Decision letter (RSOS-191429.R1)

14-Feb-2020

Dear Dr Ropert-Coudert,

On behalf of the Editors, I am pleased to inform you that your Manuscript RSOS-191429.R1 entitled "Antarctic petrels 'on the ice rocks': wintering strategy of an Antarctic seabird" has been accepted for publication in Royal Society Open Science subject to minor revision in accordance with the referee suggestions. Please find the referees' comments at the end of this email.

The reviewers and Subject Editor have recommended publication, but also suggest some minor revisions to your manuscript. Therefore, I invite you to respond to the comments and revise your manuscript.

- Ethics statement

- Data accessibility

<http://datadryad.org/submit?journalID=RSOS&manu=RSOS-191429.R1>

- Competing interests

- Authors' contributions

AB carried out the molecular lab work, participated in data analysis, carried out sequence alignments, participated in the design of the study and drafted the manuscript; CD carried out the statistical analyses; EF collected field data; GH conceived of the study, designed the study,

coordinated the study and helped draft the manuscript. All authors gave final approval for publication.

- Acknowledgements

- Funding statement

Because the schedule for publication is very tight, it is a condition of publication that you submit the revised version of your manuscript before 23-Feb-2020. Please note that the revision deadline will expire at 00.00am on this date. If you do not think you will be able to meet this date please let me know immediately.

Best regards,

on behalf of the Associate Editor, and Professor Kevin Padian (Subject Editor)
openscience@royalsociety.org

Associate Editor Comments to Author:

Thank you for so positively engaging with the reviewer's critiques - as they are largely satisfied by your responses, and recommend a number of minor additions/changes, we would like to offer you acceptance after minor revision. Congratulations, and thank you for the submission.

Reviewer comments to Author:

Reviewer: 2
Comments to the Author(s)

I believe you addressed most of my concerns and in those cases you did not address them you gave reasonable explanations.

Just a few minor suggestions:

- You refer in the response to my review that "it is important to note that we never observed any moulting feather on any of the handled birds during the breeding season, which suggests that Antarctic petrels in our study area start their moulting process later in the breeding season or after." I would suggest you include this information in page 14 line 43.

- please change the decimal symbol to dot in tables S6 onward.

Author's Response to Decision Letter for (RSOS-191429.R1)

See Appendix B.

Decision letter (RSOS-191429.R2)

17-Feb-2020

Dear Dr Ropert-Coudert,

It is a pleasure to accept your manuscript entitled "Antarctic petrels 'on the ice rocks': wintering strategy of an Antarctic seabird" in its current form for publication in Royal Society Open Science.

on behalf of the Associate Editor, and Professor Kevin Padian (Subject Editor)
openscience@royalsociety.org

Centre d'Études
Biologiques de
Chizé

Appendix A

Villiers-en-Bois, jeudi 19 mars 2020

OBJECT: Response to reviewers Manuscript ID RSOS-191429

Yan ROPERT-COUDERT

Directeur de Recherche
T. (33) 05 49 09 35 11
yan.ropert-coudert@cebc.cnrs.fr

Dear Madam, or Sir,

Thank you very much for giving us the opportunity to respond to the comments of the reviewers on our manuscript (RSOS-191429). Please find below our detailed answers. We hope that you will find that we have addressed all of their concerns and that our manuscript will be deemed acceptable for publication.

Yours Sincerely,

Yan Ropert-Coudert
On behalf of Karine Delord, main author

SCAR Life Sciences Chief Officer
Standing Committee on the Antarctic Treaty System
PEW Fellow 2017

Responses to Reviewer 1

I think the authors made a good job responding all the concerns and revising the manuscript.

Thank you!

line 63: "tacking data" - repeated twice

Corrected. Thank you for spotting this.

lines 81-84 - the two sentences are somehow repeating

Indeed, the information has been condensed into one sentence.

line 204: "good indicator or prey" - should be good indicator OF prey

Corrected. Thank you for spotting this one too.

Responses to Reviewer 2

It is an interesting paper that describes the non-breeding distribution and isotopic values of Antarctic petrels. Although the paper has potential, but I believe the authors need to improve it to achieve its full potential. Below I describe some suggestions that could help the authors improve their paper.

Thank you for all your suggestions. We have tried and accommodate them all.

Abstract:

Please begin the abstract with a sentence explain why this study is important or which gap of knowledge do you want to fill in with your manuscript.

Done.

Line 31: It is not clear from your abstract (nor title) the importance of the stable isotope analysis in your study. If you just want to determine where are Antarctic petrel during the non-breeding period and their association with sea-ice concentration and icebergs, which extra information does stable isotope data give you? Explain it in the abstract.

The stable isotopes were used for estimating not only the distribution ($\delta^{13}\text{C}$) but also the dietary preferences ($\delta^{15}\text{N}$) of petrels. We have modified the abstract to reflect this.

Line 38: You did not include anything regarding to the use of the activity data in the abstract, however it is an important part of your manuscript. I would suggest you mention it in the abstract.

Yes, words restrictions had forced us to make choices in what was developed in the abstract. Activity was implicitly mentioned at the end of the abstract when we say that the birds may use icebergs as roosting places. In the revised version, we have tried and give more substance to this by adding a sentence on the activity data.

Introduction

Lines 62-64: Did you compare the non-breeding area and isotopic values between the two study years at intraindividual level?

Furthermore, if you want to check the consistency of the foraging ecology and how non-breeding distribution at individual level you should have more than 2 year-data.

Although we deployed gls in two years, we did not intend to test formally the consistency of the foraging ecology at the intra-individual level. We just compared the two years at the inter-individual level. See also the answer below about sample size.

Lines 63-64: tracking data is repeated twice

Corrected. Thank you for spotting this.

Material and methods

Sample sizes are not clearly explained.

Geolocators: You mention at the end of the introduction that you used 2-years of individual longitudinal tracking and isotopic data so you should have for each (or to the majority) of the individuals 2 years of geolocation and of stable isotopic data. You mention in methods that you deployed geolocators to 86 petrels and recovered 69 geolocators. How many individuals correspond this 69 geolocators? Do these geolocators have only 1 year of data or is there any with 2 years? So, did you track approximately 35 individuals for 2 years? How many geolocators of Biotrack and Lotek did you recovered from each year? You only used Biotrack geolocators for the activity data analysis, but I did not find a clear information of how many geolocation trips of each year you used for this analysis.

Sorry, this was indeed confusing. We have now tried to make it clearer in the text. The confusion came from the fact that some birds (16) were tracked two seasons in a row. So, the grand total of birds used that yielded usable data is 48 individuals, 64 birds-devices. The Methods and Results have been corrected to reflect this.

In terms of device type, we deployed 40 biotrack in 2013 and recovered 28 in 2013 from which 23 yielded workable data.

We deployed 30 and 20 Lotek in 2012 and 2013, respectively. We recovered 25 and 16 in 2013 and 2014, respectively. All Lotek devices worked.
The information was added to the Results.

Feather sampling: based on the figure of stable isotope results, you have feathers from 16 individuals in one year and 19 from another year. So, in total you would have at maximum 16 individual with isotopic data from the 2 study years, correct? Why don't you have more feather samples since you recovered 64 geolocators with data?

Feathers were unfortunately not systematically collected from all GLS birds in the field. In addition, some individuals were recaptured only two years after GLS deployment, which then provided us with feather samples only for the last year of GLS-tracking.

Line 78: Did you sample the feathers during the incubation period? If you collected them during the chick-rearing did you select those feathers more worn to ensure that the feathers sampled were moulted in the previous year?

Feathers were collected both during incubation or early chick rearing. As opposed to remiges/rectrices as well as belly/chest contour feathers for example, we never observed any clear pattern of wearing out on the back feathers that we collected. Therefore, we could not specifically target worn-out feathers. However, it is important to note that we never observed any moulting feather on any of the handled birds during the breeding season, which suggests that Antarctic petrels in our study area start their moulting process later in the breeding season or after.

Line 88: Do you know the breeding success of your tracked individual? It is known that migratory phenology can differ between individuals with different breeding success, with failed breeders migrating earlier and, also, moulting earlier. Thus, failed breeders could moult a greater proportion of their body feathers still during the breeding period in contrast with successful breeders that may moult the majority already in their non-breeding areas.

This a good point. However, based on our direct observations, we found absolutely no indication of the occurrence of moulting during the part of breeding season when we were collecting the feather samples. We are very confident that this did not impact our sampling and results.

Lines92-96: Maybe you should mention that although Biotrack geolocation activity data have more resolution, you do not know the sequence or duration of the wet and dry periods during those 10-min. Please refer how many geolocators did you used for this analysis (and how many from different individuals)

We used data from 23 devices that came from 23 individuals, all sampled in 2013. This should be clearer now in the revised version. We have also added a note concerning the absence of information on the sequence and exact durations of dry/wet periods in the 10-min records.

Lines 96-98: Normally petrels' behaviour is influenced by the moonlight specially the ones relying on prey such as myctophids or krill that perform the diel vertical migration. Furthermore, during the winter in such latitudes the diel vertical migration can be influenced by the moonlight. So, I would suggest you check it the moonlight influence in petrels' activity in more detailed. (1) Please considered to use a GAMM similar for example to <https://www.int-res.com/abstracts/esr/v40/p189-206/> instead of the GLMM to test the effect of the moon. In that paper the authors used a continuous variable of moon-phase instead of a categorical one. (2) Also considered to use the time spend on water instead of out of water, as I would expect that the relationship with moonlight would be greater in the time spent on water during night/dawn/dusk, because if there is moonlight birds can spend more time on water foraging (3) instead of categorized day or night use a continuous daylight variable or at least use 4 categories: day, night, dawn and dusk, because many seabirds forage during dawn and dusk and in the present manuscript is not clear if you include those periods in the daylight or in the night period.

These are interesting questions to test but we feel it may detract us from the main emphasis of our paper which is not to study at-sea night activity, but rather the association with icebergs. But following the suggestion of the referee, we keep this idea in mind for another study.

Lines102-103: please refer the units of the residence time of large icebergs

Added.

Any particular reason why you used 30% and 70% UD kernels and not the most common ones of 50% and 95% generally representing the core-area and home-range distributions?

The choice of threshold for representing kernel-based Utilization Distributions is always somewhat arbitrary. We felt

that the classical 95% UD was too large and not informative enough but we have now produced a new figure with the 30-50-95% UD as a trade-off between our preferred UD and those suggested by the referee.

Wouldn't be possible to use the directly the time spent per cell grid in the GLMM and GAMM instead of converting it to presence/absence grids?

GLMM have been performed on the distance to the edge and not on presence absence. We actually tried both GLMM and GAMMs in an earlier version, and the results were similar except for SSH (see below). We kept the presence-absence grids as the models performed much better on these (higher deviance explained ~27% for presence/absence against 2% for time spent).

Results

Lines 115-116: please refer to the Table S2 and S3 at end of this sentence.

Done.

Lines 119-120: to try to understand if the bird was flight or resting on ice you could (1) check the duration of the continuously dry periods; large periods probably correspond to resting, while short periods probably correspond to flights, although this will be different among species if there is information about the maximum continuous flight of Antarctic petrels all dry periods above that thresholds would probably correspond

to resting; (2) check in which period of the day there are long dry periods occur. In figure 2b you present the date pooled for each month making difficult to better understand the behaviour of your study species. It would be better if you check how the activity of Antarctic petrel change throughout the day (and maybe day and month) by plotting a GAMM like for example the Figure 1 of the article Ramos R., Ramírez I., Paiva V.H., Militão T., Biscoito M., Menezes D., Phillips R.A., Zino F., González-Solís J. (2016) Global spatial ecology of three closely-related gadfly petrels. *Scientific Reports* 6: 23447. Furthermore, you could calculate a grid of the “night flight index” (which in the case of your species the dry periods would be flight/resting) and compare using a GLMM with the grid of the small icebergs to check if near small icebergs your birds tend to spend more time on dry, which could suggest they are resting on those areas.

We do agree with the referee that a proper analysis of the activity data would be very interesting but this would deter us from the main goal of the present study, which is to investigate the association of petrels with icebergs and not to study the nocturnal activity while at sea. This could form the basis for another study (see also above).

Figure 2c (page 30): please include the units of the variables in the graphs or in the legend of the figure. Furthermore, in the legend you need to explain that the variable of small iceberg is representing the size of small icebergs until 3km of length, while the large iceberg is not the size but the residence time of large icebergs (>5km of length).

The information regarding the icebergs was added to the caption. Other units (SIC and SSH) were added to the caption too.

As far as I notice, you only refer in the legend of Figure 2 and not in anywhere else of the paper that the activity data is only from 2013. Please refer it in methods or in results.

Correct, we thought that the message was given on lines 93-96 in the original version, where we said that only Biotrack data were used but because we had tried to save on space for words we cut out the number of devices deployed on each year. This would have been clear then given that Biotrack were only deployed in 2013. We have now clarified this.

Are isotopic values of these potential prey available in bibliography? If there is, it would be better if you perform isotopic diet reconstruction

instead of putting here the name of the potential prey that you did not verify.

+

Figure 3 (page 31): Are there isotopic values of the three main prey species of Antarctic petrel? If there is please perform an isotopic diet reconstruction, it is more precise than just put these lines and it will allow you to know (and discuss) the proportion of krill versus fish/squid.

The isotopic values of *Euphausia superba*, *Electrona antarctica* and *Psychroteuthis glacialis* do exist in the scientific literature. However, while the $\delta^{15}\text{N}$ values of the two latter species are consistent over time and space, krill values vary a lot (in the range 2 to 8 ‰), depending on the sampling location. The lack of knowledge of $\delta^{15}\text{N}$ values of krill in the wintering area of Antarctic petrels thus precludes looking at a reconstructed diet with confidence.

Our goal was not to detail the winter diet of Antarctic petrels, but, instead to indicate that birds are likely to feed on the same prey items during both the breeding and inter-breeding period, with some inter-individual variations (as highlighted on Figure 3).

You should refer that the reference isotopic values of penguins you used are from chicks. Isotopic data from chicks may be biased regarding to adults due to metabolic process related with quick body growth. However, penguin adults moult while fasting which can also affect isotopic values. It would be better if you could find a reference isotopic values of the prey or a petrel species. If none of these not available, I agree it is better to include those of penguin feathers as you did.

We indeed used the best available reference values as measured in penguin chicks in the literature. We have now added the fact that these are measured on chicks in the caption of figure 3, and added the caution to be taken when referring to chicks' values in the supplementary material.

Table S4: These are not deltaAIC, they are probably AIC values. Please rectify and present the AICweight too.

Sorry this is indeed supposed to be AIC (not deltaAIC). However, this table is not a model selection table (AIC weight is thus not relevant) but it is simply the list of the best model per month. Yet, we now feel this table does not really bring anything to the story and as it may be confusing we prefer to remove it in the revised version.

Some of the tables of the GLMM and GAMM results are lacking the degree of freedom. Please include it in all the respective tables

Degrees of freedom have now been added to the caption of each figure.

Discussion

Line 169: Be careful as you may be contradicting yourself as in the abstract and part of the discussion you suggest that Antarctic petrels rely on myctophid fish and krill both performing diel vertical migration and probably influenced by the moonlight.

The referee is correct. We have modified the statement in the Discussion to highlight this contradiction. It now reads “Interestingly, and in contradiction with these references, the behaviour of Antarctic petrels was not affected by the lunar cycle at the fine temporal scale of our study, while the potential prey, namely myctophid fish and *E. superba*, both perform diel vertical migration and are known to be influenced by the moonlight. This contradiction requires further investigations.”

Centre d'Études
Biologiques de
Chizé

Appendix B

Villiers-en-Bois, Friday 14 February 2020

OBJECT: Response to reviewer Manuscript ID RSOS-191429

Yan **ROPERT-COUDERT**

Directeur de Recherche
T. (33) 05 49 09 35 11
yan.ropert-coudert@cebc.cnrs.fr

Dear Madam, or Sir,

Thank you very much for this nice answer to the revision of our manuscript (RSOS-191429). Please find below our detailed answers to the remaining comments of the referee. We also include the author statement.

Yours Sincerely,

Yan Ropert-Coudert
On behalf of Karine Delord, main author

SCAR Life Sciences Chief Officer
Standing Committee on the Antarctic Treaty System
PEW Fellow 2017

Responses to Reviewer

I believe you addressed most of my concerns and in those cases you did not address them you gave reasonable explanations.

Thank you!

- You refer in the response to my review that "it is important to note that we never observed any moulting feather on any of the handled birds during the breeding season, which suggests that Antarctic petrels in our study area start their moulting process later in the breeding season or after." I would suggest you include this information in page 14 line 43.

Done. The sentence is now included in the main text.

- please change the decimal symbol to dot in tables S6 onward.

Again, well spotted. This is a classical Fr-Eng mistake!

CEBC (UMR 7273)
Université de la Rochelle
79360 Villiers-en-Bois
www.cebc.cnrs.fr

Sous la co-tutelle de

Author-supplied statements

Ethics

Does your article include research that required ethical approval or permits?:

Yes

Statement (if applicable):

All capture and handling procedures were in accordance with the permits provided by the Norwegian Animal Research Authority (NARA/FDU permits 3714 and 5746).

Data

It is a condition of publication that data, code and materials supporting your paper are made publicly available. Does your paper present new data?:

Yes

Statement (if applicable):

Datasets supporting this article were uploaded as part of the electronic supplementary material. The data used in this study are also available on Movebank (movebank.org) and are published in the Movebank Data Repository with DOI:10.5441/001/1.q4gn4q56

Conflict of interest

We declare we have no competing interests

Authors' contributions

This paper has multiple authors and our individual contributions were as below:

Study design: K.D., S.D., A.T. Fieldwork: S.D., A.T., Data analysis and processing: K.D., A.K., C.C., F.O., A.T.. K.D. wrote the text and all authors edited and revised the manuscript, gave final approval for publication and agreed to be held accountable for the content therein.